# "You're still worth it": The moral and relational context of politically motivated unfriending decisions in online networks

German Neubaum[1]*, Manuel Cargnino[1], Stephan Winter[2], Shira Dvir-Gvirsman[3]

**1** Junior Research Group "Digital Citizenship in Network Technologies", University of Duisburg-Essen, Duisburg, Germany, **2** Media Psychology, University of Koblenz-Landau, Landau, Germany, **3** Department of Communication, Tel Aviv University, Tel Aviv, Israel

* german.neubaum@uni-due.de

**Data Availability Statement:** The questionnaire, stimulus material, data, and supplementary analyses are available at https://osf.io/ptgmq/?

## Abstract

Political disagreements in social media can result in removing (i.e., "unfriending") a person from one's online network. Given that such actions could lead to the (ideological) homogenization of networks, it is pivotal to understand the psychological processes intertwined in unfriending decisions. This requires not only addressing different types of disagreements but also analyzing them in the relational context they occur. This article proposes that political disagreements leading to drastic measures such as unfriending are attributable to more deeply rooted *moral* dissents. Based on moral foundations theory and relationship regulation research, this work presents empirical evidence from two experiments. In both studies, subjects rated political statements (that violated different moral foundations) with regard to perceived reprehensibility and the likelihood of unfriending the source. Study 1 ($N = 721$) revealed that moral judgments of a political statement are moderately related to the unfriending decision. Study 2 ($N = 822$) replicated this finding but indicated that unfriending is less likely when the source of the morally reprehensible statement is relationally close to the unfriender and provides emotional support. This research extends unfriending literature by pointing to morality as a new dimension of analysis and offers initial evidence uncovering the psychological trade-off behind the decision of terminating digital ties. Drawing on this, our findings inform research on the homogenization of online networks by indicating that selective avoidance (in the form of politically motivated unfriending) is conditional upon the relational context and the interpersonal benefits individuals receive therein.

## Introduction

Although social networking sites (SNS) are typically used to stay in contact with friends or to gather new acquaintances, it may also happen that some of these connections are terminated by the users. This so-called "unfriending" can occur for many reasons such as avoiding uninteresting, unimportant or inappropriate content posted by the unfriended person [e.g., 1–3]. Since SNS have also become venues of politically and civically relevant debates, research has started to focus on reasons for dissolving online ties, showing that 10–18% of users unfriended or unfollowed someone because of political disagreements [4–8].

view_only=
8967804364ab40a1b69873aab84ca9f5.

**Funding:** This research was supported by the Digital Society research program funded by the Ministry of Culture and Science of the German State of North Rhine-Westphalia (Grant Number: 005-1709-0004), Junior Research Group "Digital Citizenship in Network Technologies" (Project Number: 1706dgn009). We acknowledge support by the Open Access Publication Fund of the University of Duisburg-Essen. The funders had no role in study design, data collection and analysis, decision to publish, or preparation of the manuscript.

**Competing interests:** The authors have declared that no competing interests exist.

Investigating people's unfriending behavior in virtual networks is becoming increasingly relevant because the termination of digital connections has consequences not only on an interpersonal but also on a political level. If users tend to unfriend people when encountering political disagreements, there is a risk that their networks become ideologically homogenous, which could lead to the formation of echo chambers [9–15]. In this context, unfriending or blocking a source in one's network could be interpreted as a manifestation of political intolerance [16] in the sense of post hoc banning content (e.g., suggested news articles or user-generated comments) that challenges one's personal political viewpoints.

This research focuses on unfriending decisions that are motivated by identifying political dissents between oneself and others and argues that–in many cases–*political* disagreements are rooted in *moral* disagreements, manifesting themselves in (perceived) discrepancies of fundamental moral intuitions between the unfriender and the unfriended person [17–19]. A political statement on Facebook (FB) arguing, for instance, that it is not a good idea to implement a women's quota in companies or that it could be dangerous to welcome a large number of refugees in one country, consequently, could be interpreted as violating the moral principles of equality and charity. This perceived violation, then, could lead to the assessment of irreconcilable moral differences and, therefore, to the decision of breaking the digital bond with the source of this statement. Perceived moral violations, though, might not be perceived as equally wrong in every relational context: Previous works suggested that some expressions or actions could be judged as immoral in some relational constellations (e.g., in interactions with relationally distant ties such as acquaintances) but not in others (with relationally close ties such as family members; [18–20].

To address the complexity of unfriending decisions, it seems crucial to not only consider the moral roots of political disagreements but also the relational context in which these occur. While previous works have made explicit connections between morality and politics [19, 21, 22] as well as political disagreements and unfriending [5, 7, 23], this research offers a theoretical integration of those lines of inquiry and takes the examination of this phenomenon one step further by incorporating the interpersonal context. More specifically, we argue that moral dissents through political statements will lead to different outcomes depending on *who* is the source of the statement and *what benefits* this person has to offer. Based on two empirical studies, this research investigates (a) the moral domains in which (perceived) violations in the form of political statements lead to terminating digital ties, (b) the relational contexts in which moral violations entail unfriending decisions, and (c) the potential social rewards (i.e., social support) that might inhibit users to unfriend or unfollow someone despite moral dissents. With this, the current work offers a more nuanced theorizing of social media users' inner calculus of which factors motivate and prevent them from actively intervening in the political structure of their online network.

## Politically motivated unfriending behavior

Social media research has started to focus on the affordance of unfriending and blocking someone in one's personal digital network especially in order to find out (a) who are the unfrienders, (b) who are the unfriended, and (c) why people unfriend others [2, 24–26]. Given the contemporary discussion of echo chambers and how social media contribute to creating such like-minded cocoons online, politically motivated unfriending behavior and its conditions have been linked to the notion of users as "active homogenizers" of their social media environments [4, 5, 7, 23].

### Who are the unfrienders?

Characterizing politically motivated unfrienders, studies found that unfriending behavior is more likely among those who use social networking services more intensely and those who

have more friends on these platforms. This suggests that both factors come along with a greater likelihood to encounter political disagreements as well as a greater literacy in terms of knowing how to actively shape one's network [5, 8, 23]. Studies also found that especially those who have greater political interest, are ideologically extreme, and less supportive of free speech tend to terminate digital connections because of political disagreements [4, 5, 7, 23]. While these findings clearly suggest that any form of political involvement predicts the post hoc filtration of one's network, it seems that political orientation has no clear predictive value: There is evidence for both, left-leaning [4, 27] and right-leaning [5] users being susceptible to unfriend others due to political dissents.

## Who are the unfriended?

Concerning who is more likely to be unfriended, research consistently revealed the fragility of weak ties: When users encounter a political disagreement online, they are more likely to unfriend the source of this political statement when this person is not relationally close to the unfriender, manifested through, for instance, a lack of offline interaction [2, 5, 28]. At least two explanations may account for this consistent finding: First, relationally closer ties could be politically more similar to oneself than relationally distant ties [29]. Second, relationally distant ties are more dispensable because they do not offer the kind of social rewards that relationally closer ties offer [30].

## Why do people unfriend?

Consistently, there is a set of reasons that individuals mention when it comes to explain their political unfriending: People tend to dissolve digital relationships in a political context when they find that the unfriended publishes political content too frequently, expresses him/herself in an offensive or uncivilized manner or voices a political stance with which the unfriender disagrees [4, 5, 31]. The present work focuses on political disagreement as a motive for political unfriending and intends to uncover the psychological mechanisms at work when political dissents come up in online networks. In this regard, John and Gal [32] proposed that social networking platforms represent "personal public spheres" in which individuals do not necessarily adhere to collectively shared values of the public sphere (e.g., tolerance toward opposing views) but to norms that govern their personal sphere (e.g., deciding over one's social interactions). This interwovenness of different spheres in users' lives may consequently facilitate avoidance and detachment compared to political communication in other contexts different from social media. Similarly, Schwarz and Shani [33] trace unfriending back to the circumstance that SNS bring together individuals from different social sub-networks of an individual. These networks normally are governed by different sets of social norms (e.g., norms regulating to not discuss politics or to adhere to civil modes of communication). As these norms are blurred on SNS, such environments expose users to social information they would not have come across otherwise, eventually leading users to re-evaluate (and potentially dismiss) their online relationships. Further evidence focused on the psychological mechanisms underlying unfriending decisions in such situations: Following the concept of cognitive dissonance [34] which suggests that individuals feel uncomfortable when they encounter information contradicting their own cognitions (e.g., knowledge or beliefs), Jeong, Zo, Lee, and Ceran [35] examined the psychology behind encountering cross-cutting information online. According to their results, the more individuals use social media, the more likely they are to encounter cross-cutting viewpoints which, in turn, is positively associated with a subjective state of mental discomfort. Users try to end this psychologically uncomfortable state by actively modifying their social environments (e.g., disconnecting from those who disagree).

To conclude, on a psychological level, unfriending decisions appear to be rooted in cognitive dissonance regarding political viewpoints. However, the question arises whether the state of psychological discomfort emerges from the mere inconsistency between two political opinions or by the assessment that there is a divergence of more deeply rooted cognitions, for instance, moral intuitions. In the present work, we argue that *political* disagreements that cause unfriending behavior can be traced back to *moral* disagreements.

## The moral basis for unfriending decisions

Moral disagreements can be understood as differences in people's valuing of distinct moral foundations. According to moral foundations theory [17, 18], people's moral judgments can be structured using five different domains: (1) Harm/care as the ideal of protecting and taking care of others, (2) fairness as the principle of justice and a fair distribution of resources, (3) loyalty as the ideal of standing with the ingroup (such as family but also one's nation), (4) authority as the ideal of respecting legitimate authorities, and (5) purity as the principle to avoid happenings that "contaminate" humanity or the earth. Previous research showed that people's moral foundations predict their political orientation: For instance, fairness and care positively predict a political left-wing orientation [21, 22], while loyalty, authority, and purity are related to a political orientation toward the right [30]. When exploring these moral intuitions in different societies, scholars found empirical evidence for a two-factor structure: Accordingly, the foundations harm/care and fairness are represented in one factor called "individualizing or liberal intuitions," while the foundations purity, loyalty, and authority were grouped to the factor "binding or conservative intuitions" [e.g., 29–32, 36].

According to this line of inquiry, human beings use their individual set of moral foundations to evaluate actions or expression as right or wrong [37]. This set is also used when assessing the moral nature of a political debate or its related decision: Empirical studies indicated that individuals with stronger individualizing foundations prefer a more egalitarian society, while individuals with stronger binding foundations are more inclined to see the world as dangerous, asking for more social control [38–40]. Following this logic, when users argue about political issues on SNS, discrepancies not only in their political views [4, 23] but also in their prioritization of moral foundations become identifiable [41]. While simple disagreements on political issues may not be sufficient for terminating a digital relationship, dissents in deeply rooted moral views may be.

## The relational context of moral judgments

Not every disagreement, though, provokes the same reaction [42]. The relational context of a social interaction, for instance, in terms of interpersonal closeness has been proposed to be "inherently important in understanding variability in moral judgment" [19, p. 1]. Since different relationships activate different moral motives and different standards of evaluation, what might be acceptable in one relationship, can be wrong in another [43]. Previous research showed that judgments of moral violations vary significantly across different relational settings [43, 44–47]. For instance, violations of the principle of fairness are perceived as least wrong when they happen in a relational context that is marked by communal sharing (e.g., kinships), since in these kinds of relationships people tend to take freely from each other without paying attention to an equal distribution of resources [43].

In social media communication, the kind of relationship is often defined by the type and the extent of social support users receive from an online tie [30, 48, 49]. According to the multidimensional view of social support [50], emotional support represents the extent to which ties offer a person compassion and affection, informational support refers to the extent to

which ties offer useful information (e.g., recommendations for restaurants), and instrumental support reflects the provision of tangible aid in the form of immaterial help or material resources. These types of social support are likely factors that users take into account when deciding whether to delete a person from one's online network [30]. In face of a political statement on SNS that violates a personally important moral principle, users may engage in a psychological trade-off: When unfriending this person, they would be able to reduce the immediate (and potentially future) cognitive dissonance due to the moral discrepancy. At the same time, individuals have a fundamental need to belong and to maintain interpersonal relationships, even if these relations are detrimental [51]. Dissolving a digital relationship could entail interpersonal conflicts, violate social norms, and lead the unfriender to lose different forms of social support or other benefits the unfriended provides [51, 52]. Especially close relationships have been consistently found as important sources of social support, leading to increased levels of well-being and mental health [53–55]. Therefore, individuals may be more reluctant to jeopardize these benefits and gratifications when the political statement they disagree with comes from a close tie–even if they realize, based on the political statement, that there are moral discrepancies (that induce cognitive dissonance) between them and the close friend. The present work is dedicated to address this complex inner weighing that precedes unfriending decisions in light of moral discrepancies. From a theoretical point of view, this trade-off appears to be governed by two important psychological forces: one's need to reduce states of cognitive dissonance but also the need to belong and to receive social support.

## The present research

Drawing on this theoretical background, the present work is intended to not only replicate previous findings on individual differences fostering political unfriending behavior but also point to situations in which politically motivated unfriending is more likely to occur. To this end, this work proposes a new link between political disagreements, moral judgments, and the evaluation of interpersonal benefits (such as social support) as an explanation for the outcome of an unfriending decision.

Replicating previous research on the phenomenon of unfriending [5, 7, 23], this work expects that unfriending or blocking someone due to a political disagreement is more prevalent among those who use Facebook more frequently (*Hypothesis 1*), have more Facebook friends (*Hypothesis 2*), are politically extreme (*Hypothesis 3*), and are more politically interested (*Hypothesis 4*). Taking the interpersonal context into account, this study intends to replicate previous findings on the role of relational closeness [2, 5] by hypothesizing that, given a political disagreement, SNS users are more likely to unfriend or block relationally distant compared to relationally close ties (*Hypothesis 5*).

Assuming a more nuanced view on the nature of a political disagreement, the present research draws on the notion that political debates are often accompanied by moral judgments [21, 38, 39]. This leads to the assumption that the likelihood of unfriending or blocking someone because of their political expression is associated with the extent to which the unfriender perceives this statement to be a moral violation in the sense of interpreting it as reprehensible (*Hypothesis 6*). However, given that the prioritization of moral foundations varies across individuals and societies [22, 56, 57], not all types of perceived moral violations might be perceived as equally wrong and, therefore, irreconcilable. Consequently, it is asked on an explorative level, to what extent are some moral violations in the form of political comments on SNS judged as more wrong than others (*Research Question 1*) and for which moral foundations are violations in the form of a political comment on SNS most likely to entail unfriending/blocking behavior (*Research Question 2*).

Yet, people may judge the same moral trespass differently depending on the nature of relationship to the violator [43]. Due to the lack of evidence related to the effect of relational contexts on the perceptions of moral violations in the form of political comments, it is asked whether SNS users judge a moral violation as more or less wrong depending on the relational closeness to its source (*Research Question 3*). It seems conceivable that the relational context affects the likelihood of unfriending or blocking a violator of moral foundations because of the social gratifications that come with increasing relational closeness [30]. More specifically, receiving social support from an online tie might outweigh the perceived moral wrongness of a political statement and, thus, increase the wish to maintain the (virtual) relationship. In this case, receiving social support could explain why people are more reluctant to unfriend or block a person when they are a close tie. It is expected that the effect of relational closeness on the likelihood of unfriending/blocking someone is mediated by received social support from this person (*Hypothesis 7*). An overview of hypotheses and research questions is given in A1 Fig in S1 File.

## Study 1

### Objectives

The objectives of Study 1 were to replicate previous findings about who is more likely to unfriend or block someone because of political disagreements (*H1-H4*) and provide initial evidence for the connection between moral evaluations of a political statement and the decision of unfriending someone (*H6, RQ1, and RQ2*).

### Method

The questionnaire, stimulus material, data, and supplementary analyses of Study 1 are documented as S2 File.

**Sample.** A total of 721 adult Facebook users (467 female, 252 male, 2 no specification) whose age ranged from 18 to 75 (*M* = 37.66, *SD* = 13.35) were recruited by a non-commercial German online panel. Participants were invited to a study dealing with "friendships on Facebook." Most participants (91.4%) had at least a university entrance qualification and used Facebook daily (56.9%) or once or multiple times per week (23.9%). No incentive was given to participants. More information about participants' demographics in comparison to German Facebook/SNS users is displayed in A2 and A3 Figs in S1 File. While the distribution of age groups in our sample is comparable to the distribution among German Facebook users, this comparison also reveals that there is an overrepresentation of female participants in our sample.

**Experimental design and stimulus.** This study employed a within-subjects experimental design in which each participant had to rate five different written scenarios. As stimuli, we developed 15 scenarios of someone (i.e., a non-specified Facebook friend of the participant) making a political statement that violates one of the moral domains proposed by the moral foundations theory [18]. The five scenarios each participant saw, were randomly taken from this set of 15. This procedure also ensured the randomization of presentation order for each participant. The instruction was "You are about to read five short scenarios about what people from your Facebook friends list could post. Please read carefully these scenarios and imagine a friend of yours would post something like that." Subsequently, subjects read the scenario, for example: "Imagine a Facebook friend of yours repeatedly publishes status updates on the topic of war. In these posts, this person writes that sometimes it is reasonable to wage a war, even when innocent people get harmed or killed. This is necessary to solve political conflicts." For each moral foundation, we used three political topics (see A5 Table in S1 File). The matching

of political topics with moral foundations was based on previous research connecting different moral dimensions to politically relevant issues [39]. As a manipulation check, subjects subsequently had to indicate to what extent they think this person violated each of the five moral foundations with this political statement. On a five-point scale, they had to state whether this Facebook friend ". . .endangers the security and well-being of other people" (violation of harm/care), ". . .fosters inequality and disproportionality among human beings" (violation of fairness), ". . .shows a lack of loyalty toward his/her own group (family, friends, countrymen)" (violation of loyalty), ". . .shows a lack of respect for authority" (violation of authority), or ". . . jeopardizes the purity of the world or humanity" (violation of purity). As can be seen in A6 Table in S1 File, the intended manipulations appeared to be largely successful.

Participants gave their consent through the online survey system. This procedure was approved by the ethics committee of the division of Computer Science and Applied Cognitive Sciences at the Faculty of Engineering, University of Duisburg-Essen (May 08, 2017).

**Measures.** The psychometric properties of this study's central measures are reported in A7 Table in S1 File.

*Past unfriending/blocking behavior*. Based on previous studies on political unfriending [5, 6], we asked subjects if they ever unfriended or blocked a Facebook friend and, if so, whether political reasons are responsible for this decision: "Have you ever (a) unfriended / (b) blocked a person from/of your Facebook friends list because of the following reasons?" "This person. . ." (1) "posted too frequently about politics," (2) "posted something political with which I disagreed," (3) "argued about politics with me or with someone I know," (4) "disagreed with something political I posted," (5) "posted something political that offended me or friends of mine," (6) "other reason." Participants responded to each item in a dichotomous response format (yes/no).

*Perceived wrongness*. For each of the five scenarios, subjects rated on a five-point scale to what extent they found this statement to be wrong (based on [43]).

*Likelihood of unfriending*. Participants also indicated how likely it is that they unfriend this person on Facebook because of posting such statements (on a five-point scale from 1 = very unlikely to 5 = very likely).

*Further measures*. Subjects' frequency of Facebook use was measured on a seven-point scale (1 = never to 7 = every day). They were also asked to type in how many Facebook friends they had. One item was used each for measuring political interest (1 = no interest at all to 5 = high interest) and political orientation (1 = left to 10 = right), while the latter was used to calculate political extremity by coding those at both scale end points as high extremity and the middle point as low extremity. We also assessed participants' personal moral foundations (based on the Moral Foundations Questionnaire; [46, 47]), reaching acceptable reliabilities for individualizing (12 items: Cronbach's $\alpha$ = .77) and binding foundations (18 items; Cronbach's $\alpha$ = .84).

## Results

77.4% of subjects stated that they at some point unfriended someone on Facebook. Similarly, 67.4% blocked a FB friend at least once. In terms of political reasons for this behavior, 22.3% indicated that they unfriended (23.9% blocked) someone because this person expressed something political with which they did not agree. Moreover, 8.9% unfriended (9.2% blocked) a FB friend because this person posted something political that was offensive to subjects or their friends, 2.5% unfriended (4.7% blocked) someone because this person posted too often about politics, 2.2% unfriended (1.8% blocked) a FB tie because this person disagreed with something political the subject had posted, while 0.7% unfriended (1.4% blocked) someone because the unfriended person argued about politics with the participants or someone they knew.

**Table 1. Logistic regression for previous unfriending behavior (Study 1 & Study 2).**

| | Study 1 | | | | | Study 2 | | | | |
|---|---|---|---|---|---|---|---|---|---|---|
| | | | 95% CI for Odds Ratio | | | | | 95% CI for Odds Ratio | | |
| Variables | B (SE) | Wald | Lower | Odds Ratio | Upper | B (SE) | Wald | Lower | Odds Ratio | Upper |
| Frequency of FB Use | 0.36 *** (0.10) | 13.95 | 1.19 | 1.43 | 1.73 | 0.21* (0.10) | 3.88 | 1.00 | 1.23 | 1.51 |
| Number of FB friends | 0.00 (0.00) | 2.27 | 1.00 | 1.00 | 1.00 | 0.00 (0.00) | 0.02 | 0.99 | 1.00 | 1.00 |
| Political Interest | 0.29** (0.10) | 7.92 | 1.09 | 1.34 | 1.64 | 0.21* (0.10) | 4.12 | 1.01 | 1.23 | 1.51 |
| Political Ideology | -0.16* (0.08) | 4.67 | 0.73 | 0.85 | 0.96 | -0.15** (0.05) | 9.88 | 0.79 | 0.86 | 0.95 |
| Political Extremity | -0.14 (0.12) | 1.34 | 0.69 | 0.87 | 1.10 | 0.10 (0.07) | 2.00 | 0.96 | 1.10 | 1.26 |
| Constant | -3.74 (0.92) | 16.52 | | 0.02 | | -2.43** (0.84) | 8.36 | | 0.09 | |

Note: 0 = did not unfriend because of political disagreement, 1 = unfriended someone because of political disagreement

Study 1: $N$ = 721; Study 2: $N$ = 822

***$p < .001$

**$p < .01$

*$p < .05$

*H1-H4* were tested by a logistic regression analysis considering previous unfriending behavior because of a political disagreement as dichotomous dependent variable, $\chi^2$ (5) = 38.57, $p < .001$; $R^2$ (Cox & Snell) = .05, $R^2$ (Nagelkerke) = .08. As can be seen in Table 1, results support *H1* showing that the more frequent people used FB the more likely were they to unfriend someone because of a political disagreement. Moreover, higher political interest (supporting *H4*) and a more left-leaning political orientation were positively associated with previous unfriending behavior. *H2* (i.e., the number of FB friends is positively associated with unfriending/blocking) and *H3* (i.e., political extremity is positively related to unfriending/blocking) were not supported by the data.

Since each participant viewed a total of five hypothetical moral violation scenarios, we calculated the mean for the likelihood of unfriending and for the extent to which the moral violation was perceived as wrong. Overall, subjects were moderately likely to unfriend someone across all five scenarios, $M$ = 2.73, $SD$ = 1.04, while they estimated moral violations as moderately reprehensible, $M$ = 3.38, $SD$ = 0.82. A correlation analysis (based on Spearman's rho; see Table 2) supported *H6* and indicated that–across all five scenarios–the extent to which participants found the moral violation in the form of a political statement on Facebook wrong was positively associated with the likelihood of unfriending this person. This relationship was moderate in magnitude.

**Table 2. Bivariate correlations between measures of Study 1.**

| | 1. | 2. | 3. | 4. | 5. | 6. | 7. | 8. |
|---|---|---|---|---|---|---|---|---|
| 1. Unfriending Likelihood (Mean of 5 scenarios) | - | | | | | | | |
| 2. Frequency of FB Use | -.01 | - | | | | | | |
| 3. Number of FB Friends | -.10* | .36** | - | | | | | |
| 4. Political Interest | .01 | -.04 | -.00 | - | | | | |
| 5. Political Ideology (1 = left / 10 = right) | .03 | -.04 | -.05 | -.17** | - | | | |
| 6. Political Extremity | -.12** | .03 | .06 | .19** | -.61** | - | | |
| 7. Perceived Wrongness of Message (Mean of 5 scenarios) | .69** | .01 | .01 | -.03 | .01 | -.10** | - | |
| 8. Personal Individualizing Foundation | .20** | .00 | -.00 | .08* | -.26** | .13** | .25** | - |
| 9. Personal Binding Foundation | .19** | -.00 | -.14** | -.11** | .43** | -.25** | .24** | .13** |

**$p < .01$

* $p < .05$

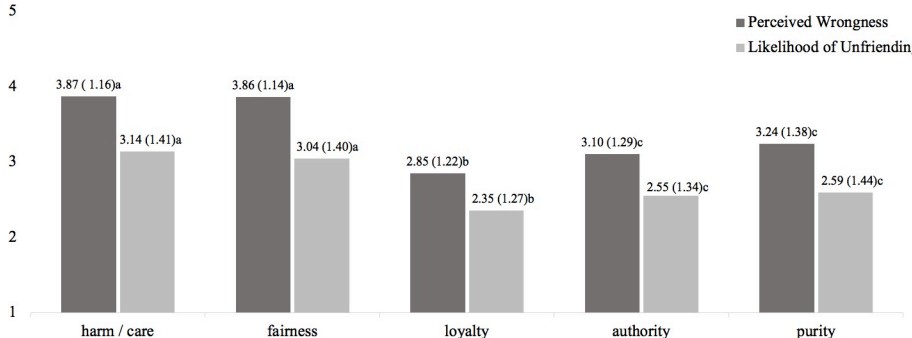

**Fig 1. Effects of scenario manipulation on perceived wrongness and unfriending likelihood in different moral domains/ means and standard deviations in parentheses (Study 1).** *Note.* Different subscripts in a row indicate significant differences with p < .05 using Bonferroni-corrected post hoc comparisons.

The correlation analysis further revealed small negative associations between likelihood of unfriending with (a) number of Facebook friends and (b) political extremity, suggesting that the more Facebook friends subjects had and the more politically extreme they were, the less likely were they to unfriend (see Table 2). Moreover, this analysis showed that the stronger participants had individualizing and binding moral foundations themselves, the more likely were they to unfriend someone who violated any moral foundation in a political statement (see Table 2). These correlations between unfriending and personal moral foundations, though, were small.

Addressing *RQ1* and *RQ2*, a MANOVA with moral foundation as independent variable and perceived wrongness as well as likelihood of unfriending as dependent variables revealed a main effect of moral foundation on perceived wrongness of the statement, $F(4,3600) = 100.10$, $p < .001$, $\eta_p^2 = .10$, and the likelihood to unfriend the person who made the statement on FB, $F(4,3600) = 44.28$, $p < .001$, $\eta_p^2 = .05$; multivariate effect: *Wilks'λ* = .90, $F(8,7198) = 49.44$, $p < .001$, $\eta_p^2 = .05$. Post hoc comparisons with Bonferroni correction showed that violations in both harm/care and fairness were perceived as equally wrong (being perceived as most wrong), followed by the domain of purity, authority, and loyalty (see Fig 1). The pattern for the likelihood of unfriending is the same: When others violate harm/care and fairness in a statement on SNS, users see the greatest likelihood to unfriend them, followed by violations in purity, authority, and loyalty (see Fig 1).

## Discussion

Study 1 presented findings that replicate previous results on who is more likely to unfriend or block someone because of a political disagreement on Facebook [5, 7], that is, those who use this SNS frequently and those who are politically interested. Moreover, Study 1 is the first one to offer a link between political statements on SNS and potential unfriending behavior by indicating that moral judgment of this statement is a driver of the unfriending decision. In a more nuanced analysis, it seems that violations in individualizing foundations are seen as more reprehensible, increasing the likelihood of unfriending relative to violations in binding foundations which may reflect the prioritizing of individualizing moral foundations in liberal western societies [22, 57].

## Study 2

### Objectives

Although Study 1 provided initial evidence on the importance of moral judgments in the face of political statements in social media, it did not take the relational context of the unfriending

decision into account. Do people remove someone who violates individualizing intuitions but to whom they have a close relationship as likely as they remove a distant tie? Is there a moral violation that is acceptable in one relationship but not in the other? Study 2 is intended to examine the interaction patterns between relationship type and moral foundation (*H5*, *H6*, *RQ1*, *RQ2*) and offer further insights into how interpersonal networks are managed online based on moral motives (*RQ3*, *H7*).

## Method

The questionnaire, stimulus material, data, and supplementary analyses of Study 2 are archived as S2 File.

**Sample.**   Participants were 822 adult Facebook users (395 female, 427 male) recruited by a German online access panel indicating that this study examines "friendships on the platform Facebook." Respondents' age ranged from 18 to 85 ($M = 43.65$, $SD = 14.46$) and, in terms of highest education degree, 34.1% finished middle school, 26.5% had a university entrance qualification, 22.9% finished college, while 14.4% had a lower educational level. Most participants used Facebook daily (56.9%) or at least once or multiple times a week (25.5%). As an incentive, participants received digital points that were equivalent to 0,75 EURO for the completion of the experiment. A comparison of the demographics of this sample and German Facebook/SNS users in general is displayed in A8 and A9 Figs in S1 File.

**Experimental design and stimulus.**   Study 2 featured a 3 (relational closeness: non-close vs. medium-close vs. close) x 2 (moral foundation of the violation: individualizing vs. binding) between-subjects experimental design. In this experiment, subjects were exposed to a written scenario in which they first had to think about a particular Facebook friend and then imagine that this Facebook friend posted a political statement which was displayed during the experiment.

The instruction related to the Facebook friend included the manipulation of relational closeness by using an adaptation of the Inclusion of Other in Self Scale [58]. In this scale, the self and another person are represented by pairs of circles varying in their distance from each other (non-close, medium-close, close). Depending on experimental condition, participants saw one pair of circles and were asked to think of their relationship to a Facebook friend which best represents the relational distance displayed by the circles. After this presentation, subjects had to indicate how much social support they received from this friend and how satisfied they are with the relationship.

To increase generalizability, each type of moral foundations was operationalized by two different political statements that violated this foundation: While the individualizing foundation was operationalized by the political topics of refugees and famine (representing the intuition care/harm), the binding foundation was operationalized by the political issues of the integrity of the national government and the police (representing the intuition authority). Each participant viewed only one political statement (shown as a Facebook status update; see https://osf.io/ptgmq/) which included a violation of the corresponding moral intuition. For instance, the violation of care/harm for the topic of refugees was: "I think, you don't compulsorily need to take refugees from war regions in Germany. Regardless of their suffering, it is not necessarily the responsibility of Germany to help out in this case" and the violation of authority for the topic of the police's integrity was: "I think, the German police is corrupt and biased. You can only despise the German police." Participants were instructed to imagine that the Facebook friend they thought of beforehand repeatedly published postings like that on his/her Facebook page.

After viewing the political statement, subjects were asked to state on a five-point scale to what extent this status update violated either individualizing (two items; Spearman Brown

coefficient = .85) or binding (three items; Cronbach's $\alpha$ = .80) foundations. A manipulation check revealed that statements about refugees and famine were perceived to violate individualizing foundations, $M$ = 3.02, $SD$ = 1.17, to a greater extent, $t(820)$ = -2.17, $p$ = .030, Cohen's $d$ = -0.15, than the statements about the government and the police, $M$ = 2.85, $SD$ = 1.09. The latter, in turn, were rated as violating binding foundations, $M$ = 2.86, $SD$ = 0.99, to a greater extent, $t(820)$ = 4.75, $p$ < .001, Cohen's $d$ = 0.33, than statements about refugees and famine, $M$ = 2.53, $SD$ = 0.99.

Participants expressed their consent through the online survey system. This procedure was approved by the ethics committee of the division of Computer Science and Applied Cognitive Sciences at the Faculty of Engineering, University of Duisburg-Essen (July 17, 2018).

**Measures.** The following measures were used in Study 2. Please note that psychometric properties of these measures are displayed in A11 Table in S1 File.

*Social support*. When participants had to think of a specific Facebook friend, they were asked to describe this person along different criteria. Based on [30] and [48], we measured social support provided by this person on three levels. Participants stated (on a five-point scale) how likely it is that the Facebook friend they thought of would provide them with (a) informational support (three items, e.g.,; "helpful information (e.g., job openings)" and "useful tips (e.g., restaurant recommendations)"; Cronbach's $\alpha$ = .85), (b) emotional support (three items, e.g., "strengthening your self-worth" and "care for you"; Cronbach's $\alpha$ = .92), and instrumental support (three items, e.g., "offering material (e.g., lending of technical devices)" and "non-material help (e.g., helping for a move)"; Cronbach's $\alpha$ = .93).

*Perceived wrongness*. Analogously to Study 1, participants were asked to state how wrong they find the stimulus political statement to be on a five-point scale from 1 = not at all reprehensible to 5 = very reprehensible.

*Likelihood of unfriending/blocking*. Participants indicated how likely it was for them to (a) unfriend and (b) block the Facebook friend for their behavior on a five-point scale (1 = very unlikely, 5 = very likely).

*Past unfriending/blocking behavior*. After being presented with the hypothetical scenario, subjects stated whether they ever unfriended/blocked someone on Facebook and if yes, whether different political reasons applied (e.g., "posted too much about politics"). The key reason this research was interested in was whether participants unfriended/blocked someone because they disagreed with the political statement the person posted on Facebook. In cases wherein this reason applied, subjects indicated the relational closeness to this person on a seven-point scale based on the Inclusion of Other in Self Scale ([48]; ranging from 1 = no overlap to 7 = most overlap). Moreover, we measured three types of social support (informational, emotional, instrumental; one item each) subjects received from the unfriended/blocked person as well as the satisfaction with the relationship (one item) with this person (both before being unfriended/blocked). Subsequently, subjects stated to what extent the political statements of the unfriended/blocked person violated certain moral intuitions on a five-point scale (five items covering individualizing and binding foundations).

*Further measures*. To address *H1-H4* and replicate previous findings on the predictors of politically motivated unfriending and blocking on Facebook, subjects indicated how frequently they used FB on a seven-point scale from 1 = never to 7 = every day as well as how many friends they had on this platform. We also measured participants' political interest (one item; 1 = no interest at all, 5 = high interest) and political orientation (one item: 0 = left, 11 = right). The latter was also recoded and used as an index of political extremity in which the middle point of political orientation represented low extremity while both scale end points reflected high extremity. Moreover, participants had also to state their personal moral foundations (based on the Moral Foundations Questionnaire; [57, 59]), with the reliabilities for

individualizing foundations (12 items), Cronbach's $\alpha$ = .83, and binding foundations (18 items), Cronbach's $\alpha$ = .84.

## Results

In terms of general prevalence of unfriending or blocking behavior, 71.2% of participants indicated that they unfriended or blocked someone on Facebook at least once. 20.4% stated that they unfriended or blocked someone because this person posted something political with which the participants disagreed. Further political reasons were: The unfriended/blocked person (a) posted something about politics that offended the participant or his/her friends (9.4%), (b) posted too frequently about politics (6.4%), (c) disagreed with something political the participant posted (3.8%), or (d) argued about politics with the participants or someone they knew (2.8%).

To address *H1-H4*, a logistic regression with previous unfriending/blocking behavior because of a political disagreement as dichotomous dependent variable was conducted, $\chi^2$ (5) = 28.01 $p < .001$; $R^2$ (Cox & Snell) = .05, $R^2$ (Nagelkerke) = .07. Results are shown in Table 1. Supporting *H1*, the analysis showed that more frequent FB users were more likely to unfriend someone because of a political disagreement. Likewise, the more politically interested (*H4*) and the more left-leaning participants' political orientation was, the higher was the likelihood for them to unfriend someone because of a political disagreement. In contrast, the number of FB friends (*H2*) and political extremity (*H3*) were no significant predictors of unfriending.

Regarding hypothetical unfriending and blocking decisions, participants, overall, indicated a medium likelihood of unfriending (*M* = 2.64, *SD* = 1.25) and blocking (*M* = 2.64, *SD* = 1.29) the Facebook friend in response to a moral violation (which in turn was perceived as moderately reprehensible, *M* = 3.20, *SD* = 1.17). *H5*, *RQ1*, and *RQ2* were tested with a MANOVA including relational closeness and moral foundation as fixed factors and perceived wrongness of moral violation, likelihood of unfriending, and blocking as dependent variables (the multivariate effects are displayed in A12 Table in S1 File). The analysis revealed a small main univariate effect of relational closeness on the likelihood of unfriending, $F(2,816) = 7.05$, $p = .001$, $\eta_p^2$ = .02, and blocking, $F(2,816) = 5.45$, $p = .004$, $\eta_p^2$ = .01. Supporting *H5*, descriptive data showed that subjects were less likely to unfriend a relationally close source of moral violation, *M* = 2.44, *SD* = 1.25, than medium-close, *M* = 2.66, *SD* = 1.20, and non-close sources, *M* = 2.84, *SD* = 1.28 (see Fig 2). The same pattern was found for blocking close, *M* = 2.46,

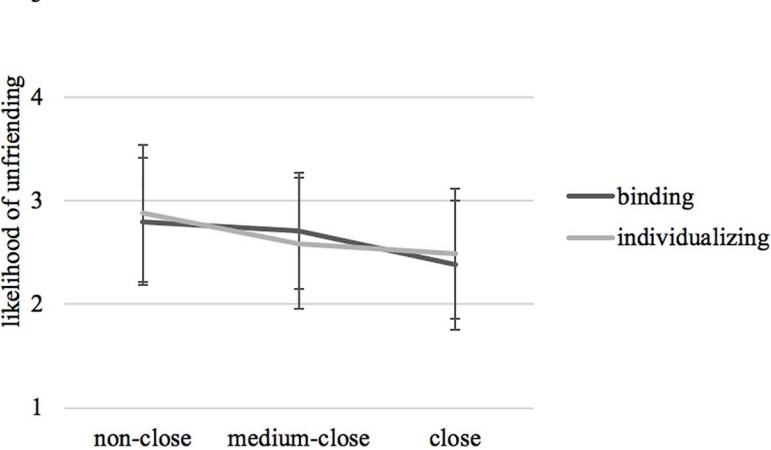

**Fig 2. Interaction effects of relational closeness and moral foundations on likelihood of unfriending.**

*SD* = 1.30, medium-close, *M* = 2.63, *SD* = 1.27, and non-close sources, *M* = 2.82, *SD* = 1.29 (see Fig 3). Post hoc comparisons with Bonferroni correction for both dependent variables revealed that the likelihood for unfriending and blocking differed significantly between close and non-close sources of moral violation, $p_{unfriend}$ = .001; $p_{block}$ = .003, while the other levels of relational closeness did not differ among each other. The findings related to *H5* are additionally corroborated by descriptive statistics related to participants past unfriending behavior: Participants who unfriended someone due to political disagreements in the past (*n* = 168) had to indicate the relational closeness to the previously unfriended based on the Inclusion into the Self Scale in which relational closeness is higher when there is a strong overlap of circles. Among the political unfrienders, 53.6% stated that there was no relational overlap between them and the unfriended, 29.2% indicated little overlap, 10.7% some overlap, 2.4% equal overlap, 2,4% very strong overlap, and 1.8% most overlap. Thus, politically motivated unfriending actions are most likely when the unfriended is a non-close tie.

As suggested by *H6*, the likelihood of unfriending and blocking someone was positively associated with the perceived wrongness of the moral violation included in the status updates. Table 3 shows that these relationships were moderate.

Focusing on *RQ1* and *RQ2*, the MANOVA did not indicate any main effects of moral foundation on perceived wrongness of moral violation, *F*(1,816) = .87, *p* = .351, $\eta_p^2$ = .00, likelihood of unfriending, *F*(1,816) = .08, *p* = .781, $\eta_p^2$ = .00, or blocking, *F*(1,816) = 1.37, *p* = .243, $\eta_p^2$ = .00, revealing that for participants it did not make a difference whether individualizing (i.e., care) or binding (i.e., authority) moral foundations were violated. Correlation analyses revealed that perceiving violations in both individualizing and binding foundations were moderately associated with unfriending and blocking this person (see Table 3). When asked about past behavior, those who unfriended or blocked someone because of a political disagreement (*n* = 168) stated that the messages that led to the unfriending/blocking decision violated individualizing foundations, *M* = 4.00, *SD* = .91, in a more emphasized manner than binding foundations, *M* = 3.21, *SD* = .90, *t*(167) = 10.55, *p* < .001, Cohen's *d* = 0.88.

Addressing *RQ3*, the MANOVA with relational closeness and moral foundation as fixed factors (see above) did not yield any interaction effects of those independent variables on perceived wrongness, *F*(2,816) = 2.34, *p* = .097, $\eta_p^2$ = .01, likelihood of unfriending, *F*(2,816) = .69, *p* = .502, $\eta_p^2$ = .00, and blocking, *F*(2,816) = .39, *p* = .675, $\eta_p^2$ = .00. In other words, the relational closeness to the expressor of a moral violation did not influence how recipients interpreted and responded to the violations of individualizing or binding foundations.

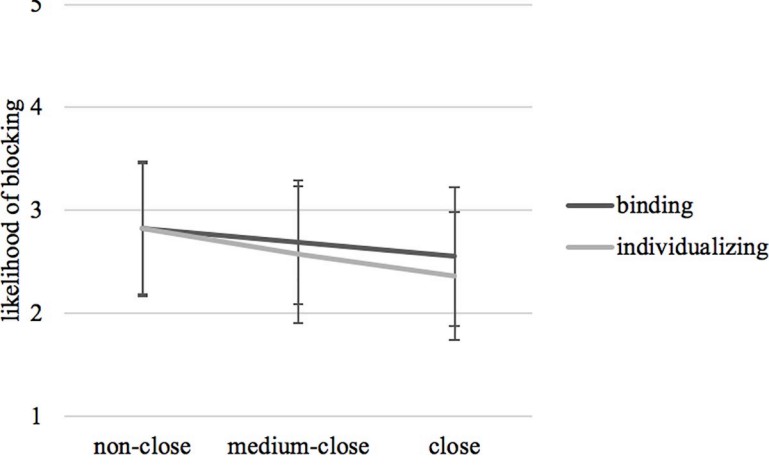

**Fig 3. Interaction effect of relational closeness and moral foundations on likelihood of blocking.**

**Table 3. Bivariate correlations between measures of Study 2.**

| | 1. | 2. | 3. | 4. | 5. | 6. | 7. | 8. | 9. | 10. | 11. | 12. | 13. | 14. | 15. |
|---|---|---|---|---|---|---|---|---|---|---|---|---|---|---|---|
| 1. Unfriending Likelihood | - | | | | | | | | | | | | | | |
| 2. Blocking Likelihood | .84** | - | | | | | | | | | | | | | |
| 3. Frequency of FB Use | -.05 | -.05 | - | | | | | | | | | | | | |
| 4. Number of FB Friends | .04 | .05 | .31** | - | | | | | | | | | | | |
| 5. Political Interest | .04 | .04 | .03 | -.03 | - | | | | | | | | | | |
| 6. Political Ideology (0 = left / 10 = right) | -.18** | -.17** | -.06 | -.02 | -.05 | - | | | | | | | | | |
| 7. Political Extremity | .00 | .00 | .06 | .08* | .24** | -.19** | - | | | | | | | | |
| 8. Perceived Wrongness of Status Update | .60** | .57** | -.03 | .10** | .08* | -.16** | -.01 | - | | | | | | | |
| 9. Perceived Violation of Individualizing Foundation | .62** | .57** | -.01 | .08* | .04 | -.16** | -.03 | .68** | - | | | | | | |
| 10. Perceived Violation of Binding Foundation | .59** | .55** | -.07* | .05 | -.03 | -.07 | -.07* | .61** | .74** | - | | | | | |
| 11. Informational Support | -.09* | -.09** | .07* | .06 | .09* | -.00 | .01 | .03 | .03 | .00 | - | | | | |
| 12. Emotional Support | -.16** | -.14** | .10** | .04 | .06 | -.00 | -.00 | -.04 | -.05 | -.05 | .78** | - | | | |
| 13. Instrumental Support | -.12** | -.11** | .07* | -.00 | .06 | -.02 | -.01 | -.02 | -.03 | -.00 | .74** | .83** | - | | |
| 14. Relationship Satisfaction | -.18** | -.16** | .11** | .02 | .01 | -.02 | .00 | -.01 | -.05 | -.10** | .61** | .74** | .69** | - | |
| 15. Personal Individualizing Foundation | .06 | .05 | .04 | -.07 | .21** | -.23** | .04 | .10** | .09** | .05 | .06 | .08* | .11** | .08* | - |
| 16. Personal Binding Foundation | -.01 | -.04 | -.05 | -.15** | .11** | .23** | -.07 | -.03 | -.01 | .15** | .15** | .12** | .15** | .05 | .37** |

**$p < .01$

*$p < .05$

When it comes to the interplay between unfriending and blocking someone and the social support received by this person, we tested *H7* (contrasting non-close and close sources of moral violations as only those differed among each other; see above) based on a structural equation model (using the software R, package lavaan by [60]). The model yielded a good fit, $\chi^2$ (42) = 99.85, $p < .001$, CFI = .99, TLI = .99, RMSEA = .05 (90% confidence interval from .04 to .06), SRMR = .02. All types of social support (included as latent variables) moderately increased with the relational closeness to a person (see A13 Table in S1 File), Informational support: $\beta$ = .57, $b$ = 1.28, 95% CI [1.11, 1.45], $z$ = 14.87, $p < .001$, emotional support: $\beta$ = .60, $b$ = 1.50, 95% CI [1.32,1.68], $z$ = 16.20, $p < .001$, instrumental support: $\beta$ = .58, $b$ = 1.40, 95% CI [1.21,1.58], $z$ = 14.96 $p < .001$. However, only a lack of emotional support made unfriending a person more likely: $\beta$ = -.41, $b$ = -.42, 95% CI [-.79, -.06], $z$ = -2.26, $p$ = .024. The indirect effect through emotional support to unfriending likelihood was significant but small (see Table 4). While the same pattern was observed for the blocking likelihood, the indirect path

**Table 4. Indirect and total effects of relational closeness on unfriending/blocking (via perceived social support).**

| | β | b | $SE_b$ | 95% CI [lower / upper] | z | p |
|---|---|---|---|---|---|---|
| *Indirect effects* | | | | | | |
| Closeness -> Informational -> Unfriending | .14 | 0.36 | .20 | -.04 / .75 | 1.79 | .074 |
| Closeness -> Emotional-> Unfriending | -.25 | -0.63 | .28 | -1.18 / -.08 | -2.25 | .025 |
| Closeness -> Instrumental-> Unfriending | .02 | 0.05 | .21 | -.36 / .45 | 0.24 | .814 |
| Closeness -> Informational -> Blocking | .09 | 0.24 | .21 | -.18 / .65 | 1.11 | .266 |
| Closeness -> Emotional-> Blocking | -.16 | -0.41 | .29 | -.98 / .15 | -1.43 | .152 |
| Closeness -> Instrumental-> Blocking | -.01 | -0.02 | .21 | -.42 / .39 | -0.08 | .939 |
| *Total effects* | | | | | | |
| Unfriending | .87 | 2.69 | .23 | 2.24 / 3.14 | 11.68 | < .001 |
| Blocking | .43 | 1.04 | .15 | .74 / 1.34 | 6.87 | < .001 |

did not reach significance (see Table 4). Thus, *H7* is only supported for emotional support. The role of receiving social support (or not) when making unfriending decisions is corroborated by descriptive statistics focusing on participants' past unfriending/blocking behavior: According to subjects who unfriended/blocked someone due to political disagreements (*n* = 168), the unfriended person never (54.2%) or seldomly (22.6%) provided informational support, never (57.7%) or seldomly (21.4%) offered emotional support, and never (69%) or seldomly (14.3%) provided instrumental support.

## Discussion

Besides corroborating moral judgment as a missing link between encountering a political statement and deciding to unfriend or block someone, Study 2 also showed that, when confronted with a posting on social media that violates fundamental moral values, SNS users are more reluctant to unfriend or block ties to whom they have a close relationship. This seems to be, at least to a certain extent, due to the higher amounts of perceived emotional support that is provided by relationally closer ties.

## General discussion

The present studies focused on the phenomenon of politically motivated unfriending as a consequence of a moral disagreement. To this end, it examined (a) whether moral judgments are at work when reading political statements on social networking platforms, (b) whether and which moral evaluations have predictive value for the decision to politically "filtrate" one's online network, and (c) to what extent moral judgements of political statements are undertaken differently depending on the relational context and the social rewards the source of political statement offers.

Study 1 and 2 partly replicated previous findings on characteristics of who are political unfrienders: Those who use social networking sites (in this case: Facebook) more frequently might encounter political dissents among their network more often and, therefore, they are more likely to unfriend or block someone [5, 8, 23]. Likewise, those who are more interested in politics feel more involved in certain political issues and, thus, perceive political disagreements as more severe, leading them to unfriend or block the dissenter [7]. What is remarkable is that, in both studies, those who lean left were more inclined to unfriend someone after a political disagreement than those who lean right. The fact that this result was not obtained consistently in previous research [4, 5, 27] may be explained by the national context and prevailing issues that are predominantly discussed in online networks at the time the study was conducted. It seems conceivable that in Germany (the country this research was conducted in 2017 and 2018), the debate about the immigration of refugees–as a rather dominant topic in the public [61]–might have led left-leaning users to unfriend or block those ties who opposed the immigration. Moreover, in contrast to previous findings [5, 23], higher political extremity was not associated with the likelihood of unfriending someone. This additionally suggests that the predictive value of the political identity or orientation on decisions of network filtrations depends on the current political landscape and state of polarization in certain countries.

While the present studies replicated certain findings from previous research, these factors only explained a limited variance of unfriending behavior (approx. 5% in both studies). In contrast to this, the present approach of focusing on the moral and relational nature of an unfriending decision appears to have greater explanatory value: First, both studies offered compelling evidence that evaluating a political statement on SNS as morally wrong is a driver of the decision to unfriend or block someone. This finding supports the notion that political disagreements that lead to the post hoc modification of one's online network are rooted in

moral discrepancies. Following this logic, Study 1 showed that individuals are able to differentiate between two different sets of moral foundations that a political statement on SNS violates: When the foundations of care/harm and fairness are trespassed, individuals seem to be slightly more inclined to unfriend someone than when the moral domains of loyalty, authority, or purity are violated. While Study 2 did not indicate main effects of the type of moral foundations violated, it showed that perceiving trespasses of individualizing foundations were slightly stronger associated with unfriending than perceived violations of binding foundations. Likewise, considering past behavior, users reported that violations of individualizing foundations were more likely to lead to unfriending decisions than trespasses of binding foundations. Collectively, this evidence is indicative for the role that morality and its nuances play when it comes to shaping one's online network by terminating a digital relationship. In line with previous research, it seems that the stronger adherence to individualizing foundations in Western countries [22, 57] relative to binding foundations is also reflected in the way individuals deal with political statements in their online network. Given the patterns observed in the present research, the question arises whether there is a likelihood that online networks become clustered by moral foundations, potentially leading to a disconnection between those who prioritize individualizing versus those who prioritize binding foundations. While previous empirical research did not corroborate the notion that social media users are captivated in politically like-minded cocoons [9–11], it seems worthwhile to assume a more complex view and examine whether sub-networks in online communication can be characterized by homogenous moral values. As suggested by Greene [62], the investigation of moral conflicts and their related emotional tensions could contribute to explaining modern tribalism, potentially also in online environments.

The threat that online communication leads to full moral clustering, though, appears rather unlikely as is suggested by Study 2 when the relational context is taken into account. In line with previous works [2, 5], in disregard of the moral violation, people were more unlikely to unfriend or block someone when this person was relationally close (compared to relationally distant). The findings also offer an explanation why: Since relationally closer ties are more likely to offer emotional support [30], individuals seem to be willing to tolerate moral violations and not terminate the digital tie, in return of maintaining the reception of social support. This result clearly shows the boundaries of the predictive value of a moral judgment in relation to a political statement and indicates the importance of a relational context for consideration. This research, thus, represents a first step to theorize the inner trade-off individuals go through when making a decision about digital interpersonal relationships in the face of political debates. In a nutshell, a specific type of social support, that is, emotional support, seems to be an inhibitor of terminating a digital connection when exposed to a political statement that violates a moral foundation. In light of the vivid debate on users' active homogenization of their online networks in terms of "echo chambers" [14, 15], this novel theoretical link between political disagreements, moral judgments, and interpersonal context in contemporary communication technologies reveals that there are boundaries to users' selective avoidance of dissents. These boundaries seem to come into play when individuals can gain something from certain relationships (i.e., network ties). A psychological view on users' decision to unfriend or block someone offers a fruitful ground for the discussion about why political and probably also moral diversity emerges and prevails in individuals' online networks [10, 15, 41].

The present findings need to be interpreted in light of this research's limitations. First, although this work assessed individuals' past unfriending and blocking behavior, it predominantly employed a scenario-based approach relying on hypothetical unfriending decisions. Both types of measures, though, seem to offer concurring findings: Those who indicated that they had unfriended because of a political disagreement in the past were significantly more likely to express a higher likelihood of unfriending in the hypothetical scenarios (on a five-

point scale/ Study 1: $M$ = 2.96, $SD$ = 0.90; Study 2: $M$ = 3.27, $SD$ = 1.20) than those who did not unfriend because of political reasons in the past (Study 1: $M$ = 2.67, $SD$ = 1.07; Study 2: $M$ = 2.53, $SD$ = 1.25) (Study 1: $t(301.62)$ = -3.49, $p$ = .001, Cohen's $d$ = -.29 / Study 2: $t(583)$ = -6.58, $p < .001$, Cohen's $d$ = -.60). Participants' answers on previous politically motivated unfriending actions also corroborated the relative importance of individualizing moral foundations, the role of relational closeness and social support. Thus, both studies reveal a significant connection between what participants told they did in the past and what they would do. Objective observations on how online networks and their potentially moral clusters change over time or mobile experience sampling questionnaires on users' smartphones, though, would be an informative complement to the present research.

Second, while the present work speculates about the motivation behind unfriending decisions, the psychological processes at work are still to be uncovered. For instance, it is unclear to what extent unfriending decisions are driven by cognitive or by affective processes. Given that moral conflicts are often fueled by emotions [62], it seems plausible to assume that not every unfriending decision is the result of a rational calculus of costs and benefits when dissolving this relationship. Future research could address the psychological mechanisms behind the creation and modification of one's online political network in a systematic manner.

Third, the composition of our samples should be taken into account. The fact that female and left-leaning participants (see A4 and A10 Figs in S1 File) were overrepresented in Study 1 leads to the question of how these variables influence unfriending decisions. While the gender-balanced sample of Study 2 indicates that politically motivated unfriending does not occur more often among women than among men, $\chi 2$ (1) = .61, $p$ = .434, both studies revealed that left-leaning individuals (compared to right-leaning ones) are more likely to unfriend others. It seems worthwhile to scrutinize whether a certain political ideology comes with enhanced involvement in certain topics which, in turn, could lead to less tolerance (and a higher unfriending likelihood) when disagreements come up or whether this finding was due to the specific national context and selected topics.

To conclude, the present work extends previous research by providing initial evidence for the importance of morality as a link between encountering politically challenging content online and actively banning this kind of information from one's news feed. If users estimate another user's political comment to be morally wrong, they will be more likely to terminate the digital relationship with this person. The power of morality, though, is limited when users are aware of the social resources, i.e., emotional support, that the person that is potentially to be unfriended can offer. This research, thus, presents a new level of analysis in online networks that could contribute (a) to understand the potential of social media communication to foster (moral) tribalism and (b) to identify the limits of this potential moral segregation in light of the social benefits human beings provide to one another.

## Supporting information

**S1 File.**
(PDF)

**S2 File.**
(TXT)

## Author Contributions

**Conceptualization:** German Neubaum, Manuel Cargnino, Stephan Winter, Shira Dvir-Gvirsman.

**Data curation:** German Neubaum.

**Formal analysis:** German Neubaum, Manuel Cargnino.

**Funding acquisition:** German Neubaum.

**Methodology:** German Neubaum, Manuel Cargnino, Stephan Winter, Shira Dvir-Gvirsman.

**Project administration:** German Neubaum.

**Writing – original draft:** German Neubaum.

**Writing – review & editing:** Manuel Cargnino, Stephan Winter.

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
