## [Decision Letter · Decision Letter 0]

1 Jun 2020

PONE-D-20-00871

“You’re still worth it” The moral and relational context of politically motivated unfriending decisions in online networks

PLOS ONE

Dear Dr. Neubaum,

Thank you for submitting your manuscript to PLOS ONE. After careful consideration, we feel that it has merit but does not fully meet PLOS ONE’s publication criteria as it currently stands. Therefore, we invite you to submit a revised version of the manuscript that addresses the points raised during the review process. 

We look forward to receiving your revised manuscript.

Kind regards,

Shang E. Ha, Ph.D.

Academic Editor

PLOS ONE

Journal Requirements:

3. Please include your tables as part of your main manuscript and remove the individual files. Please note that supplementary tables (should remain/ be uploaded) as separate "supporting information" files

Additional Editor Comments (if provided):

Reviewers' comments:

Reviewer's Responses to Questions

**Comments to the Author**

1. Is the manuscript technically sound, and do the data support the conclusions?

Reviewer #1: Partly

Reviewer #2: Yes

2. Has the statistical analysis been performed appropriately and rigorously? 

Reviewer #1: Yes

Reviewer #2: Yes

3. Have the authors made all data underlying the findings in their manuscript fully available?

Reviewer #1: Yes

Reviewer #2: No

4. Is the manuscript presented in an intelligible fashion and written in standard English?

Reviewer #1: Yes

Reviewer #2: Yes

5. Review Comments to the Author

Reviewer #1: This manuscript describes a project focused on the moral foundations of political unfriending behavior on social media, providing evidence from two experimental studies conducted in Germany. I find the topic to be timely and the proposed theoretical development intriguing, with a potential to contribute to the growing body of literature on disconnective behaviors on social media. Although I am in principle supportive of publication of this manuscript, I have some serious concerns about its execution, particularly with regards to the experimental design and analytical strategy.

First, although I think that experimental work is highly needed in this area, one of the key issues facing this study is its reliance on self-reported, hypothetical unfriending, rather than the actual unfriending behavior. Although I appreciate that the authors have noted this as a weakness, I still feel that there may be a long (and winding) road between “hypothetical unfriending” in survey context and the actual unfriending behavior on Facebook. Given the within-subjects design in Study 1, I can speculate about several confounding issues that could be at play, including social desirability, demand characteristics, etc.

Second, the current experimental designs do not allow us to make inferences such as “moral judgement is a crucial driver of the unfriending decision” since no other drivers of this behavior were tested, and that morality was also made very salient in the study context. Indeed, perhaps the biggest problem in the study design is a lack of meaningful control (or let’s say comparison) groups that would test the power of moral judgements vs. other factors (e.g. perceptions of offensiveness or incivility). Indeed, a sizeable number of respondents report unfriending others for reasons unrelated to political and ideological disagreement (10-20%), but rather caused by incivility or information overload. Perhaps the authors should emphasize more that their sole focus is on moral outrage as a consequence of encountering political disagreement, rather than more general politically-focused exposure on Facebook.

Third, although I am supportive of the authors theoretical agenda, I still find it little too exploratory and perhaps undertheorized. While the current data does not allow much more theoretical space, I would still try to consult few more studies on unfriending that could aid further theory development (e.g. John & Gal, 2018; Krämer, Hoffmann, & Eimler, 2015; Schwartz & Shani, 2016).

Fourth, I feel that the analytical approaches used could be clearer and perhaps utilize interaction terms between the key predictor variables and the past unfriending behaviors. I would personally use more graphs and charts to demonstrate the differences between the experimental groups, for example.

If the authors are keen to further develop their theoretical agenda, I would suggest using a different set of methods, which could provide a more in-depth understanding of the cognitive and affective processes at work. Furthermore, since most users today use a mobile app to access their Facebook profiles, I feel that the participants should be able to utilize their smartphones as mnemonic devices when answering the questions. I suggest looking into think-aloud protocols (Eveland & Dunwoody, 2000) and studies that use mobile media elicitation (Kaufmann, 2018; Robards & Lincoln, 2017).

Minor issues:

The manuscript still feels like an early draft and would benefit from a more structured and focused approach, and a greater attention to detail. For instance, there are still typos in several places in the manuscript (e.g. p. 10, “age raged from 18 to 75”) and the tables with bivariate correlations use a comma as a decimal separator, while the other use a decimal point. Since the manuscript is English, a decimal point should be used.

I am also unsure whether the hypotheses H1a-d really need to be proposed and tested, as they are not the main focus of the study. I would recommend either discarding them, or labeling them as separate hypotheses, as Facebook use and political extremism have very little semantic similarity, for example. I also feel that all RQs and hypotheses needs to be articulated more clearly and preferably presented in a visual form too, as a process model or similar.

Regarding the samples used in the studies, more information needs to be provided about the recruitment strategy, incentives and the demographics, comparing the participants to a general profile of German Facebook users. For instance, Study 1 seems to have a rather strong gender bias, so it would be good to discuss that further, especially in the light of the specific scenarios tested (and political ideology too).

References:

Eveland, W.P., Jr., & Dunwoody, S. (2000). Examining information processing on the World Wide Web using think aloud protocols. Media Psychology, 2, 219-244.

John, N. A., & Gal, N. (2018). “He’s got his own sea”: Political Facebook unfriending in the personal public sphere. International Journal of Communication, 12, 2971–2988.

Kaufmann, K. (2018). The smartphone as a snapshot of its use: Mobile media eliciation in qualitative interviews. Mobile Media & Communication, 6(2), 233-246.

Krämer, N., Hoffmann, L., & Eimler, S. (2015). Not breaking bonds on Facebook–mixed–methods research on the influence of individuals’ need to belong on ‘unfriending’ behavior on Facebook. International Journal of Developmental Science, 9(2), 61-74.

Robards, B. and Lincoln, S. 2017. Uncovering longitudinal life narratives: scrolling back on Facebook. Qualitative Research. 17(6): 715 -730. doi.org/10.1177/1468794117700707

Schwarz, O., & Shani, G. (2016). Culture in mediated interaction: Political defriending on Facebook and the limits of networks individualism. American Journal of Cultural Sociology, 4(3), 385-421.

Reviewer #2: Review on PONE-D-20-00871

I think the authors’ study is carefully designed and well-executed experiment whose research aim is clear with good flow. However, I have some concerns over the manuscript as follows,

Major concern over the main argument

First and foremost, I am not sure whether this study secures satisfactory level of novelty or breakthrough. Novel contributions in the authors’ argument seem unconvincing, at least, to me. There are several studies reporting the importance of morality in political decision-making and behaviors; and some previous works reported ‘unfriending on Facebook’ is triggered by political disagreements. Of course, the authors aim to try link the two sets of studies, but I am not sure the authors’ efforts to link the two sets of findings could be accepted as theoretical breakthrough. While not exactly the same, there are some previous studies examining the effect of political thoughts or evaluations on ‘unfriending’ behaviors. While morality closely considered in this manuscript is distinguished from other predictors explaining unfriending behaviors on Facebook in former studies, I think the theoretical uniqueness may not so eminent. Five dimensions of morality, already acknowledged and widespread in the field of political decision-making and behaviors, are political enough. In order to be published, the authors have to convince people (like me) why this study takes theoretically novel step. I truly believe this manuscript is well-written with good flow, but well-written paper is not necessarily a novel study. When revising the manuscript, I do hope the authors put very persuasive reasons why the authors’ theoretical attempt should be acknowledged as novel, contributing to the advance of knowledge over the unfriending behaviors with moral reasons. The authors’ revision would be heavy because such efforts should appear in the Introduction, Literature Review, and Discussion sections (even in the Abstract section), but I believe such theoretical revision would be fruitful.

Minor points but please take it seriously when revising the manuscripts

Second, when reporting results of MANOVA, I think the authors missed to report Wilks Lamda or other equivalent statistics. Additionally I think other statistical approach might be better than MANOVA because the five moral dimensions would be correlated with each other. In other words, correlations between residuals of dependent measures would be treated with other statistical methods (e.g., SUR, SEM, or random-effect models). Given that the authors already relied on SEM when testing H4, I think this suggestion might not be so difficult for the authors to adopt.

Third, the role of closeness variable in testing H4 is not clearly to me. Basically, the authors assumed that ‘closeness’ is the exogenous cause triggering mediators (i.e., Informational, Emotional, or Instrumental support) and outcomes (i.e., unfriending or blocking behavior). However, I think such mediational mechanism does not make much sense, and the closeness would take a moderating role influencing the relationships between “Informational, Emotional, or Instrumental support” and “unfriending or blocking behavior.” As reported in Table 5, most effects were found in direct effects (by the way, I calculate direct effects by taking the difference between total effect and sum of indirect effects), meaning three mediators do not well explain the relationship between closeness and both behaviors. Instead, for instance, what about hypothesizing the effect of emotional support on unfriending would be augmented under high closeness condition?

Minor but important:

Fourth, The authors posted online supplementary material at OSF, but there would be some technical problems. For example, when I clicked Table A1 on page 11, the page was not present: “The file "Moral Unfriending Study 1 and 2 - Supplementary Analyses.pdf" stored on OSF Storage was deleted via the OSF. It was deleted on Fri Jan 10 12:13:02 2020 UTC.” Other links are similar. Please check the status and please provide the supporting materials for better understanding of the readers’ manuscript.

6. PLOS authors have the option to publish the peer review history of their article (what does this mean?). If published, this will include your full peer review and any attached files.

Reviewer #1: No

Reviewer #2: No

---

## [Author Response · Author response to Decision Letter 0]

18 Sep 2020

PONE-D-20-00871: Response to the Reviewers

Dear Editor, dear Reviewers,

Before we give a detailed response to the reviewers’ comments and report on the changes we have made to the manuscript, we would like to thank you for the helpful suggestions and comments on our paper. Please let us emphasize that we really appreciate the high level of constructiveness that characterized all your suggestions. We tried to take up all feedback and feel that the manuscript indeed improved considerably. Most importantly, with the new version of the manuscript, we were able to refine the theoretical contribution of our work, outline the benefits of our methodological approach, and improve the presentation of our results. Below you find an outline of the revisions we have made to the manuscript in accordance with your comments.

R1.0:

This manuscript describes a project focused on the moral foundations of political unfriending behavior on social media, providing evidence from two experimental studies conducted in Germany. I find the topic to be timely and the proposed theoretical development intriguing, with a potential to contribute to the growing body of literature on disconnective behaviors on social media. Although I am in principle supportive of publication of this manuscript, I have some serious concerns about its execution, particularly with regards to the experimental design and analytical strategy.

Authors’ response:

We thank the reviewer for the positive assessment. We highly appreciate your constructive comments, which we tried to fully address in our revision.

R1.1:

First, although I think that experimental work is highly needed in this area, one of the key issues facing this study is its reliance on self-reported, hypothetical unfriending, rather than the actual unfriending behavior. Although I appreciate that the authors have noted this as a weakness, I still feel that there may be a long (and winding) road between “hypothetical unfriending” in survey context and the actual unfriending behavior on Facebook. Given the within-subjects design in Study 1, I can speculate about several confounding issues that could be at play, including social desirability, demand characteristics, etc.

Authors’ response:

This is a very good point that we extensively discussed among the authors when designing these studies. Indeed, as we state in our limitations section, hypothetical unfriending behavior may not fully reflect actual unfriending actions. However, there are three points that support the notion that our findings referring to hypothetical unfriending behavior are strongly indicative of actual unfriending behavior.

First, following empirical evidence related to the well-established theory of planned behavior, the intention to show a behavior is the strongest predictor of future behavior. Empirical research documented statistical relationships between intention and real behavior ranging from r = .44 to r = .47 (Armitage & Conner, 2001). Thus, we may assume that if participants expressed the intention to unfriend a person under certain circumstances (those specified in our experiment), there is a high likelihood that this person will act accordingly if faced with such a situation in real life.

Second, the connection between actual past behavior and hypothetical behavior is supported by the data of both, Study 1 and Study 2. As shown by t-tests, those who indicated that they had unfriended because of a political disagreement in the past were significantly more likely to express a higher likelihood of unfriending in the scenarios we presented (on a five-point scale/ Study 1: M = 2.96, SD = 0.90; Study 2: M = 3.27, SD = 1.20) than those who did not unfriend because of political reasons in the past (Study 1: M = 2.67, SD = 1.07; Study 2: M = 2.53, SD = 1.20) (Study 1: t(301.62) = -3.49, p = .001, Cohen’s d = -.29 / Study 2: t(583) = -6.58, p < .001, Cohen’s d = -.60). Both studies, thus, reveal a significant connection between what participants told they did in the past and what they would do. 

Third, the results on hypothetical unfriending with the focus on the moral background and social support as inhibitor are in line with the descriptive data regarding past unfriending behavior. In Study 1, we found that - in face of hypothetical scenarios - people are more likely to unfriend someone because of a violation of individualizing moral intuitions (compared to violations of binding moral intuitions). This is exactly what descriptive data of Study 2 on actual unfriending behavior in the past additionally corroborate (p. 22): “When asked about past behavior, those who unfriended or blocked someone because of a political disagreement (n = 168) stated that the messages that led to the unfriending/blocking decision violated individualizing foundations, M = 4.00, SD = .91, to a larger extent than binding foundations, M = 3.21, SD = .90, t(167) = 10.55, p < .001, Cohen’s d = 0.88.”

In Study 2, we found that - when confronted with a hypothetical scenario - participants would be more likely to unfriend a person (regardless of the moral nature of that person’s previous statement) the lower the relational closeness to this person and the less social support this person offers. Again, this is what the descriptive data on past unfriending behavior confirms: Participants who unfriended someone due to political disagreements in the past (n = 168) had to indicate the relational closeness to the previously unfriended based on the Inclusion into the Self Scale in which relational closeness is higher when there is a strong overlap of circles. Among the political unfrienders, 53.6% stated that there was no relational overlap between them and the unfriended, 29.2% indicated little overlap, 10.7% some overlap, 2.4% equal overlap, 2,4% very strong overlap, and 1.8% most overlap. Thus, politically motivated unfriending actions are most likely when the unfriended is a non-close tie. Moreover, when it comes to social support, the unfriended person never (54.2%) or seldomly (22.6%) provided informational support, never (57.7%) or seldomly (21.4%) offered emotional support, and never (69%) or seldomly (14.3%) provided instrumental support.

Taking all of these points combined into account, we believe that costs of using hypothetical scenarios in terms of a reduced ecological validity are somewhat evenly balanced with its benefits in the sense of exerting full experimental control over the moral background of statements and the relational closeness of such situations. We believe that our results are to a considerable extent reflective of what real unfriending behavior outside an experimental setting looks like. 

Given the consistency between past and hypothetical unfriending behavior in our findings, we also believe that participants’ responses did not suffer from social desirability or demand characteristics. Please note that in Study 1 (employing a within-subjects design) we only manipulated the moral intuitions that were in the center of the political status updates. Moral intuitions are considered as very strong and stable moral beliefs that vary from individual to individual (Haidt, 2012) and we do not believe that there is a “common sense” of what responses might be socially desirable when a different set of moral intuitions are presented in a row. Besides this, for each participant, the survey software randomly took 5 out of 15 possible moral violation scenarios, varying the order of statements displayed to participants in the within-subjects design, too. So, the order was different for every participant.

To address the reviewer’s point in our manuscript, we (1) included more information about the randomization of statements in the within-subjects experiment (Study 1; p. 12), (2) added our descriptive results concerning actual previous unfriending behavior that had not been included previously (see pp. 21-23), and (3) extended the limitations section by not only acknowledging that the hypothetical approach has its limits but also explaining which results indicate that there is a connection between actual and hypothetical unfriending behavior (see p. 27). 

R1.2:

Second, the current experimental designs do not allow us to make inferences such as “moral judgement is a crucial driver of the unfriending decision” since no other drivers of this behavior were tested, and that morality was also made very salient in the study context. Indeed, perhaps the biggest problem in the study design is a lack of meaningful control (or let’s say comparison) groups that would test the power of moral judgements vs. other factors (e.g. perceptions of offensiveness or incivility). Indeed, a sizeable number of respondents report unfriending others for reasons unrelated to political and ideological disagreement (10-20%), but rather caused by incivility or information overload. Perhaps the authors should emphasize more that their sole focus is on moral outrage as a consequence of encountering political disagreement, rather than more general politically-focused exposure on Facebook.

Authors’ response:

This is a fair point and we apologize if our statements were misleading. Our motivation to focus on political disagreement and its roots was triggered by the fact that in the area of politics, unfriending actions are very likely when individuals encounter a political disagreement with others. This has been corroborated by the descriptive data in John & Dvir-Gvirsman (2015) and Study 1 and 2 of the present work. In those studies, political disagreement was even the most frequent reason for politically motivated unfriending. The reviewer is right, though, that there are further reasons for unfriending in a political context such as encountering content that is offensive or someone posting too often about politics. It seems likely that in real life, all of these reasons overlap, for instance: A person repeatedly publishes political statements on Facebook that contradict the ideal of taking care of others (moral violation) using a very uncivil language. We definitely see the benefits of both, approaches that (a) analyze the unique effect of each of these reasons and (b) investigate the interaction effect of these different reasons (e.g., does a moral violation lead to unfriending only when it is expressed in an offensive manner?). In the present study, we clearly decided to pursue (a), that is, to isolate the effect of a political disagreement and uncover its psychological background. 

As suggested by the reviewer, the revised version of the manuscript is more explicit in describing the focus of our research as well as how and why we zoomed in on this phenomenon (see changes in the introduction, p. 3, 4, literature review, pp. 5-6, and in the general discussion, p. 24). The reviewer is right that we cannot make any statements on the relative importance of moral judgments as preceding unfriending decisions – we therefore solely focused on the connection between morality and political disagreements.

R1.3:

Third, although I am supportive of the authors theoretical agenda, I still find it little too exploratory and perhaps undertheorized. While the current data does not allow much more theoretical space, I would still try to consult few more studies on unfriending that could aid further theory development (e.g. John & Gal, 2018; Krämer, Hoffmann, & Eimler, 2015; Schwartz & Shani, 2016).

Authors’ response:

We thank the reviewer for these recommendations. Indeed, all three references were very helpful in refining the theoretical view on our research. In particular, before referring to the cognitive dissonance account of politically motivated unfriending, we now address the normative dimensions of SNS communication and their role of norms in the filtering of opposing political views (see pp. 5-6). Furthermore, taking the work of Krämer et al. (2015) into account, we were able to shed more light into why users may refrain from virtually disconnecting themselves from other people (see p. 9), referring to individuals’ fundamental need to belong. We believe that we construed a more coherent line of argumentation proposing that the individuals’ mental trade-off could be governed by two important psychological forces: one’s need to reduce states of cognitive dissonance and the need to belong and to maintain interpersonal relationships (see p. 9). Besides this, we also included further literature on (political) unfriending (e.g., John & Agbarya, 2020; Lopez & Ovaska, 2013)

R1.4:

Fourth, I feel that the analytical approaches used could be clearer and perhaps utilize interaction terms between the key predictor variables and the past unfriending behaviors. I would personally use more graphs and charts to demonstrate the differences between the experimental groups, for example.

Authors’ response:

This is a helpful advice. We replaced Table 3 by a graph that visually depicts the mean differences between the experimental groups (see Figure 1, p. 38). Moreover, we included new graphs representing the MANOVA calculated to address H2, RQ1, and RQ2. These graphs now more clearly depict the main effect of relational closeness and the (non-significant) interaction effect of relational closeness and moral foundations (see Figure 2 and 3; pp. 39-40).

R1.5:

If the authors are keen to further develop their theoretical agenda, I would suggest using a different set of methods, which could provide a more in-depth understanding of the cognitive and affective processes at work. Furthermore, since most users today use a mobile app to access their Facebook profiles, I feel that the participants should be able to utilize their smartphones as mnemonic devices when answering the questions. I suggest looking into think-aloud protocols (Eveland & Dunwoody, 2000) and studies that use mobile media elicitation (Kaufmann, 2018; Robards & Lincoln, 2017).

Authors’ response:

We highly acknowledge R1’s point on the need to apply methods that allow for a more in-depth understanding of how individuals react to political disagreement and at what point they form the decision to detach from a social media contact. In fact, the two studies presented here are part of a larger research project that is intended to uncover the cognitive and affective mechanisms involved when users form and intervene in their political networks. Another study from this larger project (that is in a second review cycle at another journal) employed a qualitative approach with thinking aloud protocols. This qualitative study indeed showed that the immediate responses to political disagreements often occur on an affective level in terms of anger or perceived hostility. These affective responses entail cognitive processes that try to reduce the state of cognitive dissonance such as discrediting the source of the political statement or (as researched in-depth in the present two studies) identifying moral discrepancies between oneself and the source of the political statement.

We fully agree with the reviewer’s suggestion that open-ended approaches such as thinking aloud protocols will enable us to cover the full range of responses related to encountering political disagreement; still, we believe that such an approach is outside the scope of the present article which has a very specific focus on the morality of the political disagreement and this morality is evaluated in face of varying relationship types (another factor that came up in the qualitative study). We therefore believe that the present work complements these qualitative findings yet adding the rigid and straight-forward testing of hypotheses by employing experimental methodology and large and heterogeneous samples.

We are open to add the results of that qualitative study to our literature review if the Editors and Reviewers agree to add a study that is not published yet.

Either way, we additionally mentioned the use of “mobile experience sampling questionnaires on users’ smartphones” (p. 27) as a rather promising method for future research to identify immediate responses to political disagreements.

R1.6:

Minor issues:

The manuscript still feels like an early draft and would benefit from a more structured and focused approach, and a greater attention to detail. For instance, there are still typos in several places in the manuscript (e.g. p. 10, “age raged from 18 to 75”) and the tables with bivariate correlations use a comma as a decimal separator, while the other use a decimal point. Since the manuscript is English, a decimal point should be used.

Authors’ responses:

Thank you for raising our attention in this regard. Before submitting the revision, we thoroughly went through the papers and corrected the points you mentioned as well as further errors included in the previous version.

R1.7:

I am also unsure whether the hypotheses H1a-d really need to be proposed and tested, as they are not the main focus of the study. I would recommend either discarding them, or labeling them as separate hypotheses, as Facebook use and political extremism have very little semantic similarity, for example. I also feel that all RQs and hypotheses needs to be articulated more clearly and preferably presented in a visual form too, as a process model or similar.

Authors’ responses:

With proposing H1a-d, we intended to replicate previous findings from the rather limited political unfriending research. The reviewer is right that corroborating previous findings on who is more likely to unfriend is not the main focus of our study. Still, we believe that testing these relationships in two different studies not only prevents this field from having a replication problem (as was the case for many different phenomena in psychology; Nosek et al., 2015; Open Science Collaboration, 2015), but also indicates the relative explanatory value of rather stable interindividual variables (e.g., political ideology or political interest) in contrast to situational variables such as the moral nature of a statement or the relational context. As we outline in the discussion, having evidence that connects to previous research (by intending to replicate them) vis-à-vis with a new focus on situational factors, helps us to understand the phenomenon of political unfriending in a more comprehensive way. This is why we decided to keep these hypotheses, especially because they were part of our original set of assumptions. 

To address the reviewer’s point and to increase clarity, we re-labeled them and treat them as more separate assumptions changing from H1a-d to H1-H4. Moreover, we created a new figure to visualize our set of hypotheses and research questions. Since our manuscript already included a high number of Tables and Figures, we decided to integrate the visualization of hypotheses and research questions into our supplementary material (Figure A1). If the Editor and Authors wish to have this figure in the main manuscript, we will be glad to include it there.

R1.8:

Regarding the samples used in the studies, more information needs to be provided about the recruitment strategy, incentives and the demographics, comparing the participants to a general profile of German Facebook users. For instance, Study 1 seems to have a rather strong gender bias, so it would be good to discuss that further, especially in the light of the specific scenarios tested (and political ideology too).

Authors’ response:

We gladly provide more information about our samples. Both samples were recruited through online access panels that consist of volunteers of social science research. While no incentive was given to participants in Study 1, in Study 2, participants received so-called “mingle points,” that is, reward points participants can collect in order to receive a payment (after a certain amount has been collected). The incentive for our study was 0,75 EURO. We included more information about recruiting and incentives in our manuscript (see p. 11 and pp. 16-17). Moreover, we added new Figures to our supplementary material that compare the demographics of German Facebook users in 2017 and 2018 (the years these studies were conducted) to the demographic structure of our sample. The figures reveal that in terms of age, both samples are somewhat representative for German Facebook users as the deviations in each age group are not larger than 5%. For gender, it is evident that there is an overrepresentation of female participants in Study 1 while the gender distribution in Study 2 can be seen as representative for German Facebook users. We also integrated Figures about the political ideology of participants into the supplementary material (see Figures A4 and A10). Unfortunately, we do not have any data at hand how representative these samples are in terms of political ideology concerning German Facebook users. 

Nevertheless, following the reviewer’s suggestion, we elaborate upon the limitations of our samples in the general discussion, acknowledging that in Study 1 there is an overrepresentation of female and left-leaning participants. As we state in the limitations section (pp. 27-28), it seems that there is no gender-based difference in unfriending behavior; still, we believe that political ideology could be an interesting variable to further examine. Both studies revealed a connection between ideology and unfriending – it could be that left-leaning individuals are more engaged in certain issues and, therefore, less tolerant of disagreements in certain contexts. 

R2.0:

I think the authors’ study is carefully designed and well-executed experiment whose research aim is clear with good flow. However, I have some concerns over the manuscript as follows,

Authors’ response:

First, let us thank you for your constructive comments to improve our work.

R2.1:

Major concern over the main argument

First and foremost, I am not sure whether this study secures satisfactory level of novelty or breakthrough. Novel contributions in the authors’ argument seem unconvincing, at least, to me. There are several studies reporting the importance of morality in political decision-making and behaviors; and some previous works reported ‘unfriending on Facebook’ is triggered by political disagreements. Of course, the authors aim to try link the two sets of studies, but I am not sure the authors’ efforts to link the two sets of findings could be accepted as theoretical breakthrough. While not exactly the same, there are some previous studies examining the effect of political thoughts or evaluations on ‘unfriending’ behaviors. While morality closely considered in this manuscript is distinguished from other predictors explaining unfriending behaviors on Facebook in former studies, I think the theoretical uniqueness may not so eminent. Five dimensions of morality, already acknowledged and widespread in the field of political decision-making and behaviors, are political enough. In order to be published, the authors have to convince people (like me) why this study takes theoretically novel step. I truly believe this manuscript is well-written with good flow, but well-written paper is not necessarily a novel study. When revising the manuscript, I do hope the authors put very persuasive reasons why the authors’ theoretical attempt should be acknowledged as novel, contributing to the advance of knowledge over the unfriending behaviors with moral reasons. The authors’ revision would be heavy because such efforts should appear in the Introduction, Literature Review, and Discussion sections (even in the Abstract section), but I believe such theoretical revision would be fruitful.

Authors’ response:

Thank you so much for this thought-provoking comment that (combined with Reviewer 1’s point R1.3) helped us to refine our focus and the novel contribution of this piece. The reviewer is right that there is already convincing evidence presenting the link between morality and politics. This is a point that we certainly recognized, and we are sorry that our novel claim was not explicit enough in the previous version of the manuscript. Our line of argumentation draws on the connection between morality and politics as a fundamental premise but goes one step further by bringing the role of interpersonal relationships into play. First, from a psychological point of view, research on the connection between morality, moral congruence and interpersonal relationships is indeed scarce (Simpson et al., 2016; Simpson & Laham, 2015). Second, and more importantly, we believe that in the unique context of social media this triangle of morality, politics, and interpersonal relationships is key to understand the psychological processes that govern the formation and evolution of online social networks. When a person befriends or unfriends someone, it has consequences for the structure of a network. If morality is an important factor guiding the decision to unfriend someone, there is a theoretical risk that – in the long run – online social networks become not only politically but also morally homogeneous. Providing insights into different psychological levels of an individual’s decision to unfriend someone, therefore, takes the general political unfriending literature one theoretical step further but also contributes to the vivid debate of “echo chambers” and “filter bubbles,” indicating new dimensions on which online networks can become uniform.

In a nutshell, we believe that the present research (a) offers more in-depth views into the psychological black box behind politically motivated unfriending decisions, (b) presents innovative theoretical explanations – based on morality and its role in interpersonal relationships – for the formation of homogeneous communication clusters, and (c) provides initial empirical evidence for the limits of a morality-based homogenization, for instance, through the fact that people consider a variety of benefits (not only reducing cognitive dissonance, but also receiving social support) when it comes to make unfriending decisions.

In fact, the reviewers’ comments helped us to become more explicit in our research motivation and contributed, in our view, to outline the novelty of our line of thinking in a more convincing manner. As suggested by the reviewer, we elaborated on this novel strep in greater detail in the abstract (see p. 2), introduction (see p. 4), theoretical section (see p. 9), and discussion (see p. 26). We hope that our novel contribution becomes more evident to the reader now.

R2.2:

Minor points but please take it seriously when revising the manuscripts

Second, when reporting results of MANOVA, I think the authors missed to report Wilks Lamda or other equivalent statistics. Additionally I think other statistical approach might be better than MANOVA because the five moral dimensions would be correlated with each other. In other words, correlations between residuals of dependent measures would be treated with other statistical methods (e.g., SUR, SEM, or random-effect models). Given that the authors already relied on SEM when testing H4, I think this suggestion might not be so difficult for the authors to adopt.

Authors’ response:

We are sorry for making misleading statements. The reviewer is right that while we report the multivariate effects for Study 2 (see Table A12), we missed reporting it for the MANOVA in Study 1. We added this value to the manuscript (see p. 15). It seems to us that our statements were not specific enough as we did NOT use moral intuitions as dependent variables. For both MANOVAS, we used “perceived wrongness” and “likelihood of unfriending/blocking” as DVs. In our understanding, it is advisable to run a MANOVA when dependent variables are correlated and could represent one group (Field, 2013). We know from correlation analyses (Table 2 and 3) that these variables are moderately correlated, and it is advisable to not ignore this kind of relationships among dependent variables (Field, 2013). We decided to run MANOVAS specifically for that reason. In the revised version of the manuscript, we tried to be more specific about the variables that were used as independent and dependent variables in our analyses (see p. 15). If the reviewer had a different point in mind, we are happy to include other analyses to the manuscript.

R2.3:

Third, the role of closeness variable in testing H4 is not clearly to me. Basically, the authors assumed that ‘closeness’ is the exogenous cause triggering mediators (i.e., Informational, Emotional, or Instrumental support) and outcomes (i.e., unfriending or blocking behavior). However, I think such mediational mechanism does not make much sense, and the closeness would take a moderating role influencing the relationships between “Informational, Emotional, or Instrumental support” and “unfriending or blocking behavior.” As reported in Table 5, most effects were found in direct effects (by the way, I calculate direct effects by taking the difference between total effect and sum of indirect effects), meaning three mediators do not well explain the relationship between closeness and both behaviors. Instead, for instance, what about hypothesizing the effect of emotional support on unfriending would be augmented under high closeness condition?

Authors’ response:

This is an interesting point that provoked discussions within the group of the authors. We really went back to which research question we were originally interested in. While a moderation analysis can answer the question “WHEN does relational closeness lower the likelihood of unfriending or blocking,” a mediation analysis addresses the question “WHY does relational closeness reduce the likelihood of unfriending or blocking.” Psychological literature consistently showed that people receive stronger social support in relationally close relationships compared to relationally distant ones (Feeney & Collins, 2015; Lakey et al., 2014; Sarason & Sarason, 2006); thus, we need to expect a strong effect of relational closeness on different types of social support (which, in fact, is corroborated by our results). Given this line of thinking and the literature providing evidence for this strong connection between closeness and social support, it is clear to us that social support can only serve as the explanation WHY relational closeness reduces the likelihood of unfriending and blocking and not WHEN. Even if we were interested in the question of “WHEN,” the compelling evidence documenting the connection between relational closeness and social support would suggest that commonly there should be only little variation of social support within the different types of relational closeness. For instance, we will find only a small number of participants who receive low levels of social support in close relationships. This would lead to an unfair comparison between the different groups of relational closeness. 

The reviewer is right that the indirect effect is not very strong. But following the open science movement and the ideal that hypotheses should be presented as they were originally conceptualized (even if results do not reveal strong effects; Kerr, 1998; Nosek et al., 2018)), we decided keep our hypothesis as it was originally proposed. However, we extended the introduction of this hypothesis to justify the idea behind a mediation and to make our line of reasoning more transparent (see p. 9 and pp. 10-11). 

R2.4:

Minor but important:

Fourth, The authors posted online supplementary material at OSF, but there would be some technical problems. For example, when I clicked Table A1 on page 11, the page was not present: “The file "Moral Unfriending Study 1 and 2 - Supplementary Analyses.pdf" stored on OSF Storage was deleted via the OSF. It was deleted on Fri Jan 10 12:13:02 2020 UTC.” Other links are similar. Please check the status and please provide the supporting materials for better understanding of the readers’ manuscript.

Authors’ response:

We apologize that we missed to update the URLs. We did so in the last stage of this revision so that the links should work now.

Again, thank you for your suggestions!

All the best,

The authors

References

Armitage, C. J., & Conner, M. (2001). Efficacy of the Theory of Planned Behaviour: A meta-analytic review. British Journal of Social Psychology, 40(4), 471–499. https://doi.org/10.1348/014466601164939

Bode, L. (2016). Pruning the news feed: Unfriending and unfollowing political content on social media. Research & Politics, 3(3), 205316801666187. https://doi.org/10.1177/2053168016661873

Feeney, B. C., & Collins, N. L. (2015). A New Look at Social Support: A Theoretical Perspective on Thriving Through Relationships. Personality and Social Psychology Review, 19(2), 113–147. https://doi.org/10.1177/1088868314544222

Field, A. (2013). Discovering statistics using IBM SPSS statistics. sage.

Haidt, J. (2012). The righteous mind: Why good people are divided by politics and religion. Vintage.

John, N. A., & Dvir-Gvirsman, S. (2015). “I Don’t Like You Any More”: Facebook Unfriending by Israelis During the Israel-Gaza Conflict of 2014: Facebook Unfriending in Israel-Gaza Conflict 2014. Journal of Communication, 65(6), 953–974. https://doi.org/10.1111/jcom.12188

John, N., & Agbarya, A. (2020). Punching up or turning away? Palestinians unfriending Jewish Israelis on Facebook. New Media & Society, 146144482090825. https://doi.org/10.1177/1461444820908256

Kerr, N. L. (1998). HARKing: Hypothesizing After the Results are Known. Personality and Social Psychology Review, 2(3), 196–217. https://doi.org/10.1207/s15327957pspr0203_4

Lakey, B., Cooper, C., Cronin, A., & Whitaker, T. (2014). Symbolic providers help people regulate affect relationally: Implications for perceived support: Symbolic providers regulate affect. Personal Relationships, 21(3), 404–419. https://doi.org/10.1111/pere.12038

Lopez, M. G., & Ovaska, S. (2013). A look at unsociability on Facebook. 27th International BCS Human Computer Interaction Conference (HCI 2013) 27, 1–10.

Nosek, B. A., Alter, G., Banks, G. C., Borsboom, D., Bowman, S. D., Breckler, S. J., Buck, S., Chambers, C. D., Chin, G., Christensen, G., Contestabile, M., Dafoe, A., Eich, E., Freese, J., Glennerster, R., Goroff, D., Green, D. P., Hesse, B., Humphreys, M., … Yarkoni, T. (2015). Promoting an open research culture. Science, 348(6242), 1422–1425. https://doi.org/10.1126/science.aab2374

Nosek, Brian A., Ebersole, C. R., DeHaven, A. C., & Mellor, D. T. (2018). The preregistration revolution. Proceedings of the National Academy of Sciences, 115(11), 2600–2606. https://doi.org/10.1073/pnas.1708274114

Open Science Collaboration. (2015). Estimating the reproducibility of psychological science. Science, 349(6251), aac4716–aac4716. https://doi.org/10.1126/science.aac4716

Sarason, B. R., & Sarason, I. G. (2006). Close Relationships and Social Support: Implications for the Measurement of Social Support. In A. L. Vangelisti & D. Perlman (Eds.), The Cambridge Handbook of Personal Relationships (pp. 429–444). Cambridge University Press. https://doi.org/10.1017/CBO9780511606632.024

Simpson, A., & Laham, S. M. (2015). Different relational models underlie prototypical left and right positions on social issues: Relational models, ideology, and social issues. European Journal of Social Psychology, 45(2), 204–217. https://doi.org/10.1002/ejsp.2074

Simpson, A., Laham, S. M., & Fiske, A. P. (2016). Wrongness in different relationships: Relational context effects on moral judgment. The Journal of Social Psychology, 156(6), 594–609. https://doi.org/10.1080/00224545.2016.1140118

---

## [Decision Letter · Decision Letter 1]

16 Nov 2020

“You’re still worth it” The moral and relational context of politically motivated unfriending decisions in online networks

PONE-D-20-00871R1

Dear Dr. Neubaum,

We’re pleased to inform you that your manuscript has been judged scientifically suitable for publication and will be formally accepted for publication once it meets all outstanding technical requirements.

Kind regards,

Shang E. Ha, Ph.D.

Academic Editor

PLOS ONE

Additional Editor Comments (optional):

Reviewers' comments:

Reviewer's Responses to Questions

**Comments to the Author**

1. If the authors have adequately addressed your comments raised in a previous round of review and you feel that this manuscript is now acceptable for publication, you may indicate that here to bypass the “Comments to the Author” section, enter your conflict of interest statement in the “Confidential to Editor” section, and submit your "Accept" recommendation.

Reviewer #1: All comments have been addressed

Reviewer #2: All comments have been addressed

2. Is the manuscript technically sound, and do the data support the conclusions?

Reviewer #1: Yes

Reviewer #2: Partly

3. Has the statistical analysis been performed appropriately and rigorously? 

Reviewer #1: Yes

Reviewer #2: Yes

4. Have the authors made all data underlying the findings in their manuscript fully available?

Reviewer #1: Yes

Reviewer #2: Yes

5. Is the manuscript presented in an intelligible fashion and written in standard English?

Reviewer #1: Yes

Reviewer #2: Yes

6. Review Comments to the Author

Reviewer #1: In general, I am quite satisfied with the revisions that the authors made to their manuscript in line with the reviewers comments. I still feel that the quality of writing could be improved a bit and that the research could be presented more succinctly. Related to that, I would personally discard H1-H4 and focus more on the other hypotheses and RQs. Since I know that the authors are keen on presenting the findings H1-H4, I suggest that they present them in a more descriptive form, without posing them as hypotheses and that their discussion is minimized. The contribution of this research should be more clearly stated in the discussion, preferably with some "big picture" view on the political implications of disconnection on social media.

Reviewer #2: I found the revised manuscript is improved much better. While I have still a slight doubt over the novelty of the manuscript (see the first point in the previous round of review), I think the manuscript is a well-written paper with some noticeable findings. As a final word, I would like to suggest that the authors put clearer reasons why and how their study is discriminated from previous studies dealing with the similar, even though not identical, topics. I am unsure whether another round of review can remove the slight doubt in my mind; thus I would like to leave the final decision to the editors.

7. PLOS authors have the option to publish the peer review history of their article (what does this mean?). If published, this will include your full peer review and any attached files.

Reviewer #1: No

Reviewer #2: No

---

## [Editor Report · Acceptance letter]

26 Nov 2020

PONE-D-20-00871R1 

“You’re still worth it” The moral and relational context of politically motivated unfriending decisions in online networks 

Dear Dr. Neubaum:

I'm pleased to inform you that your manuscript has been deemed suitable for publication in PLOS ONE. Congratulations! Your manuscript is now with our production department. 

Kind regards, 

on behalf of

Dr. Shang E. Ha 

Academic Editor

PLOS ONE